# SingMonitor: E-bike Charging Health Monitoring Using Sound from Power Supplies

Xiangyong Jian, Lanqing Yang, Yijie Li, Yi-Chao Chen and Guangtao Xue *

School of Electronic Information and Electrical Engineering, Shanghai Jiao Tong University,
Shanghai 200000, China
* Correspondence: xue-gt@cs.sjtu.edu.cn

**Abstract:** In recent years, fire disasters caused by charging electric bicycles/moped (e-bikes) have been increasing, causing catastrophic loss of life and property; Worse still, existing fire warning systems are costly to install and maintain, and they work after the accident occurs. Some existing works propose using power meters or similar sensors in the power grid to monitor e-bike charging health. However, the use of additional equipment makes them challenging to deploy. Others can use the sound or electromagnetic signals emitted by e-bikes for monitoring, but they suffer from limited monitoring distance. To solve this problem, we propose *SingMonitor*, a scheme to remotely monitor e-bike charging status using mobile devices' microphones. The charging e-bike generates a unique current signal, which is then transmitted through the power grid and drives the mobile devices' power supply to generate sound, which is then captured by a microphone. Based on this principle and the proposed template matching method, *SingMonitor* can identify the e-bike charging status. Experiments show *SingMonitor* achieves an F1 score of 0.94 in identifying 10 e-bikes' charging status, with a detection distance of 9m+.

**Keywords:** e-bike; power supply; power factor correction; template matching





## 1. Introduction

In recent years, electric bicycles/electric mopeds (here, both are referred to as "e-bikes") have become increasingly popular, corresponding with growing environmental awareness and technological advances in the e-bike industry, with the global e-bike market reaching USD 17.83 billion in 2021 [1]. Although e-bikes are convenient, they also provide some safety threats. For example, there were nearly 18,000 fire disasters caused by charging e-bikes in China in 2021 [2]. Similarly, in October 2022, the USA recalled about 22,000 e-bikes, whose lithium batteries could ignite, explode, or spark, posing fire, explosion, and burn hazards to consumers [3]. Low-quality batteries or chargers are some of the main reasons for these disasters [4,5]. Furthermore, in Section 3.1, we show how charger or battery quality issues can influence a battery's capacity and even increase the risk of fire while charging. Worse yet, over time and with improper use (charging immediately after parking, etc.), even high-quality chargers and batteries can develop issues and become unreliable [6,7].

To prevent these disasters, many systems are dedicated to fire detection and alert, which can significantly reduce the damage caused by e-bike fires. For example, the Fire Alarm Control Unit (FACU) [8] is the mainstream commercial fire warning system. FACU consists of three parts: initiation devices (including various sensors and pull stations), the control panel, and sirens and fire suppression devices. Refs. [9–15] focus on improving FACU systems with IoT technology. Such studies need to deploy many sensors or communication nodes with integrated sensors for rapid fire detection and alarm. They differ in the hardware and communication protocols used. Refs. [16,17] analyze data captured by closed-circuit television or webcams using image processing technology to detect and track

fires. However, these methods are costly to install and maintain and only work after a fire hazard occurs.

Some appliance detection methods can also monitor the charging health of e-bikes. For example, some of them use sensors to directly obtain the operating current, voltage, and load consumption of the appliances in the circuit [18–24]. Others use side-channel methods to identify appliances and their operating status by the electromagnetic [25,26] or acoustic signals [27,28] generated when the appliance is in operation. However, most of these methods require extra hardware, which makes them difficult for large-scale deployment, and some have limited monitoring distances.

Some battery monitoring methods [29,30] use sensors to obtain information such as the voltage and current of the battery to monitor the battery. Ref. [31] trains an LSTM model with voltage, temperature, and other information to monitor and protect the battery. Ref. [32] predicts the battery status through an electrochemical model. These methods require extra hardware devices to obtain information such as current and voltage, which increases the system's cost.

To solve this problem, in this paper, we propose *SingMonitor*, a scheme to remotely monitor e-bike charging health using mobile devices' microphones. *SingMonitor* can monitor the charger or battery for faults during daily charging, slowing battery capacity loss and thus preventing potential fire hazards. We create the *SingMonitor* system based on observations that: (1) The charging e-bike can inject unique current signals into the power grid. Currently, e-bikes commonly use three-stage battery chargers [33]. At different charging stages, the power factor correction (PFC) circuit inside the e-bike charger can inject different feedback currents, i.e., PFC signals, into the power grid; (2) The PFC signals can transmit through the power grid. As shown in Figure 1, PFC signals can be propagated along wires throughout the power grid. A detailed explanation can be found in Section 3.3; (3) The PFC signals can drive the power supply at the other end of the grid to generate sounds. When the power supply of another appliance connected to the same distribution system receives this signal, it will emit a sound under electromagnetic forces; (4) and the sounds can be captured by nearby mobile devices and used to monitor the e-bike's charging health. By analyzing this sound signal, *SingMonitor* can obtain the current charging stage of the e-bike, as well as the charge duration. After comparing with the normal charging pattern (see Section 3.1), the system can determine whether there is a problem with the charger and battery, thus monitoring the e-bike's charging health.

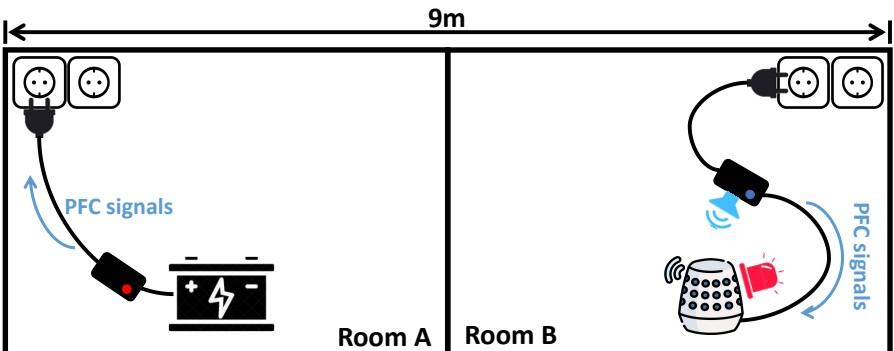

**Figure 1.** Diagram of signal propagation process. PFC signal generated by charger travels along grid and affects power supply to smart speaker, causing it to emit sound.

In developing the proposed *SingMonitor*, we encountered several challenges: (1) Interference of noises. The PFC signals are exposed to background noise interference from the power grid, environment, the power supply's internal components, and the sound acquisition equipment during propagation and conversion to sound signals. PFC signals generated by other appliances can also interfere with the e-bike's PFC signal. (2) It is difficult to distinguish different e-bikes and their charging stages in a complex appliance environment. Considering that the types, numbers, and combinations of appliances

in different homes vary, we need to design a method that can be trained with only e-bike data and still perform well in complex appliance environments to improve our system's usability. (3) It is also difficult to define a healthy charging status.

We then proposed many schemes to solve these problems: (1) We propose a noise cancellation method based on Variational Mode Decomposition (VMD), periodicity detection, and spectral subtraction. We adopted VMD and periodicity detection to eliminate the aperiodic background noise. To obtain clean e-bike charge-stage templates (see Section 5), we employed spectral subtraction to remove the interference of other appliances. (2) We proposed a template-matching-based scheme and designed a new distance metric to distinguish different e-bikes and their charging stages. We defined a new similarity metric by compared the differences in center frequency and bandwidth based on the characteristic that the center frequency and bandwidth of PFC signals generated by different appliances and e-bikes' different charging stages are different. (3) We propose to build the normal charging pattern by calculating the duration of different charging stages in the registration phase (see Section 5). The system compares the charge duration measured during the health monitoring phase with the normal charging pattern. If the gap between the two exceeds a threshold (which is dynamically adjusted), the system considers the e-bike to be in an unhealthy charging status.

Our contributions can be summarized as follows:

1. We propose and implement a system that uses sound emitted by the power supply to monitor e-bike charging health;
2. We propose a noise cancellation scheme using VMD, periodicity detection, and spectral subtraction to cancel both background noise and interference from other appliances;
3. We propose a template-matching-based scheme and designed a new distance metric to distinguish different charging stages;
4. We define healthy charging states by comparing the duration of each charging stage measured by the system with the normal charging pattern;
5. We evaluate *SingMonitor* in real-world scenarios, and our experiments show *SingMonitor* achieves an F1 score of 0.94 in identifying 10 e-bikes' charging stages, with a detection distance of 9m+.

## 2. Related Work

In this section, we review techniques in four areas relevant to this paper, i.e., fire alarm systems, electrical appliances detection, battery monitoring, and PFC and switching-mode power supply (SMPS).

### 2.1. Fire Alarm System

There are many existing studies that focus on improving fire alarm systems to prevent fire disasters. For example, refs. [9–15] improve the FACU's performance by changing the sensors and boosting communication efficiency between sensor nodes. Refs. [16,17] use image-processing technology to enable the rapid detection and treatment of fires. Ref. [34] employs a camera with an optical and thermal lens for kitchen fire warnings. However, a common problem with these methods is the high installation and maintenance cost; additionally, camera-based methods may raise privacy concerns.

### 2.2. Electrical Appliance Detection

Many studies in appliance detection can also monitor the charging health of e-bikes. We split such research into two categories based on the channel they used.

**Direct load monitoring:** Such studies install sensors directly in the circuit to obtain current and voltage information. Intrusive appliance monitoring installs sensors near each appliance. Ref. [18] integrates several types of sensors within the socket, which is used to collect the usage of nearby appliances, as well as environmental data. Ref. [19] chooses to use smart plugs. However, intrusive methods are costly due to the extensive use of sensors. There are also many methods using non-intrusive load monitoring (NALM) instead. NALM [20] typically employs only one sensor to monitor the circuit's total power

consumption. This information is utilized to analyze the power consumption pattern of individual appliances and other information, such as appliance usage time. Ref. [21] can estimate the number and energy consumption of the individual loads by analyzing the voltage and current waveform of the total load. Ref. [22] analyzes household appliances' power consumption patterns using a single sensor to obtain real power data. Refs. [23,24] employ current harmonic as the core feature to identify load types. Nevertheless, NALM is also difficult for large-scale deployment because it requires extra hardware.

**Side-channel:** Side-channel technology uses the various physical signals generated by appliances during their operation to enable appliance detection. For example, ElectriSense [25] utilizes electromagnetic interference (EMI) from the switching-mode power supply to obtain individual appliance usage information. Ref. [26] collects power-draw information using a mobile electromagnetic field (EMF) sensor. Ref. [27] recognizes appliances by the sound users make when using the appliance and the sound produced by the appliance itself. Ref. [28] correlate an appliance's inherent acoustic noise with its energy consumption pattern and collect ambient sound to obtain its energy consumption pattern. However, this method has a limited monitoring range. Besides, all these works rely on extra sensors, making them hard for large-scale deployment.

### 2.3. Battery Monitoring

Many studies focus on battery monitoring. For example, ref. [29] measures the battery's voltage and temperature by installing wireless sensors inside each cell to assess the battery's state of health. Ref. [30] designs an IoT system to obtain information such as battery voltage and upload it to the control interface to facilitate the user to know and deal with the abnormal status of the battery in time. Ref. [31] designs a battery management system using sensors to obtain parameters such as battery voltage, current, and temperature. These parameters are used to train an LSTM model to monitor and protect the battery status. Ref. [32] proposes an electrochemical model that monitors the cell state by calculating parameters such as open-circuit voltage, liquid-phase diffusion, etc. The calculation of these parameters relies on the reading of current, voltage, and other data. However, these methods require extra hardware to obtain data, such as current and voltage, which raises the system cost. Compared with these direct-style monitoring methods, We propose a side-channel method that does not require current and voltage. Also, these methods are mainly for electric vehicles or lithium batteries, which are not necessarily applicable to our target.

### 2.4. PFC and Switching-Mode Power Supply

Many other works use PFC or SMPS signals. For example, ref. [35] found that PFC signals can precisely reveal information about devices' power-draw, they implemented a power side-channel attack by utilizing this trait to infer computer application launching. Changes in the computer's power consumption can affect rhe high-frequency voltage ripple generated by its PFC circuits, and NoDE [36] exploits this phenomenon to achieve data exfiltration. Ref. [37] attacks air-gapped, audio-gapped systems by planting malware to manipulate the computer's SMPS to emit specific frequency sounds. CapSpeaker [38] employs the capacitor's inverse piezoelectric effect to issue malicious voice commands to attack nearby smart speakers. Ref. [39] uses the sound from the SMPS and other noises produced by appliances to identify different appliances and use their location for indoor localization. We are inspired by these works.

### 3. Background

In Section 1, we described how *SingMonitor* works: The charging e-bike can inject unique PFC signals into the power grid. The PFC signals are then transmitted through the power grid. The transmitted PFC signals can drive the power supply at the other end of the grid to generate sound. The sounds can be captured by nearby mobile devices and be used to monitor the charging health of e-bikes. In this section, we first discuss the causes and effects of e-bike overcharging; second, we describe why e-bikes can generate a

unique PFC signal when charging; then, we describe why this PFC signal can propagate through the grid; and finally, we describe why this PFC signal can drive the power supply to produce sound.

### 3.1. E-bike Overcharging and Its Causes

**E-bike overcharging:** E-bikes are mainly powered by lead-acid and lithium batteries, which have current market shares of 30.83% and 46.41%, respectively [40]. Overcharging can cause e-bike battery capacity loss or even fire. Overcharging for lead-acid batteries raises the internal temperature, producing harmful gases and drying out the electrolyte, ultimately leading to positive grid corrosion. Grid corrosion can significantly shorten battery life, but lead-acid batteries are generally less susceptible to fire [41]. Fires caused by overcharging typically occur on lithium batteries. Lithium batteries usually consist of a positive and negative pole material and a separator that divides the two. Overcharging can cause severe side reactions within the lithium battery, which can cause the internal separator to rupture, resulting in severe thermal runaway and, eventually, a fire [42].

**Causes of e-bike overcharging:** E-bike chargers mainly use three-stage charging: constant current, constant voltage, and float charging. Figure 2 shows that the charger indicator is red during the constant current and constant voltage stages. In the experiments, we use an ammeter to measure the change in current to distinguish between these two stages. When the battery is charged to 80~95%, the charger enters the float stage, and the indicator light turns green. The charger monitors the charging current and the battery voltage via an internal feedback circuit, which decides when to switch charging stages. When the charger charges the battery with a high current or voltage for a long time due to quality problems, it cannot switch to the float stage in time and the battery will be overcharged, resulting in a loss of capacity or even a fire. Also, when the battery is faulty and thus cannot reach its rated voltage during charging, the charger will continue to supply power at a high current, thus causing the battery to overcharge. The duration of each stage varies depending on the charger and the battery. *SingMonitor* calculates the duration of each charging stage during initialization (see Section 5) to build the normal charging pattern for the corresponding e-bike. Long-term float charging is generally harmless, and *SingMonitor* is more concerned with the duration of the constant-current and constant-voltage stages.

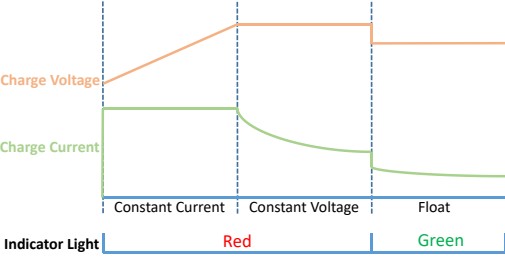

**Figure 2.** 3-stage charging.

### 3.2. E-Bike Charging Generates Unique Power Factor Correction Signals

In this section, we explain why e-bike charging can generate unique PFC signals.

**Power factor (PF):** In electrical engineering, the power factor is defined as the ratio between real power consumed by a load and the total real-plus-reactive power, which is also known as the total apparent power [43], i.e., $PF = P_{real}/P_{apparent}$. Figure 3 shows that PF can reach 1 when the input current can follow the instantaneous line voltage without distortion. This means that the total apparent power is the lowest while consuming the same real power, thus realizing the most efficient use of electrical energy. In other cases, PF is less than 1, meaning electrical energy is wasted in the distribution system.

**Power factor correction (PFC):** PFC circuits [44] can increase PF to reduce power losses by eliminating high harmonics through low-pass filters or by shaping the input current. Therefore, PFC circuits are required in many types of electrical appliances according to

the IEC61000-3-2 standard [45]. The PFC circuit in e-bike chargers is implemented by a pulse width modulator (PWM). A PWM modulates current by periodically generating high-frequency ripple to make the input current follow instantaneous line voltage. The modulation of the input current by PFC circuits injects feedback currents into the grid, i.e., PFC signals. The research [35] shows that load power consumption information can modulate the amplitude and frequency of PFC signals. The features of PFC signals can also be affected by the design and manufacturing process of PFC circuits. Therefore, as shown in Figure 4, there are obvious differences in the spectrum of the PFC signals generated by different e-bikes, appliances, and charging stages of the same e-bike. Because of the difference between PFC signals, *SingMonitor* can accurately analyze whether e-bikes are charging and which charging stage they are in from the mixed PFC signals generated by multiple appliances.

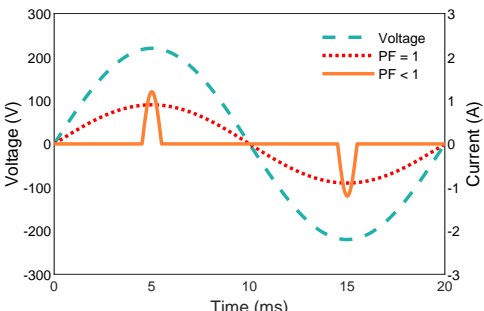

**Figure 3.** Effect of current waveform on power factor.

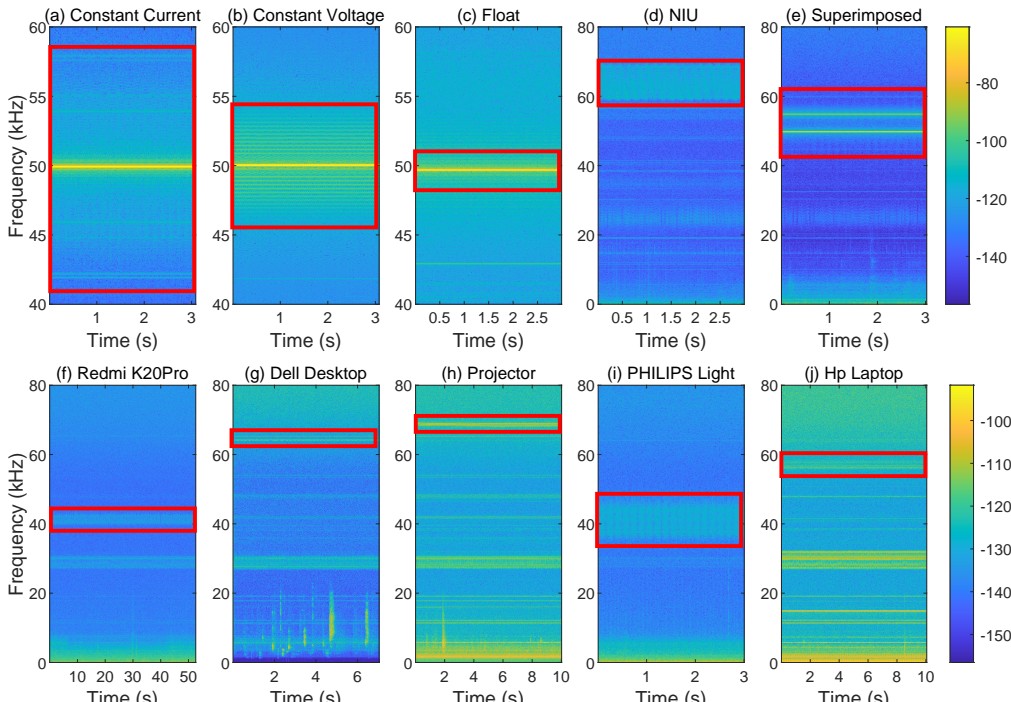

**Figure 4.** PFC signals generated by different e-bikes and appliances. (**a**–**c**) are generated by the XDAO (160 W) e-bike during different charging stages. (**d**) is generated by the NIU (160 W) e-bike during the constant current stage. (**e**) is the superimposed signal generated when YADEA (180 W) and XDAO work simultaneously. (**d**–**f**) are PFC signals generated by Redmi K20Pro (27 W), Dell desktop (360 W), Projector (225 W), PHILIPS Light (18 W), and Hp Laptop (90 W), respectively.

### 3.3. Propagation of PFC Signals in the Power Grid

This section explains how PFC signals propagate through the grid. A typical household grid consists of a distribution box and multiple distributed sockets with appliances

connected in parallel. Figure 5 is an abstract illustration of a typical household grid. For the convenience of discussion, we assume that two sockets are connected to the e-bike battery or the smart speaker. The following will explain how the current $I_0$ containing the PFC signal affects the input current $I_x$ of the smart speaker. Where $V_s$ denotes the source voltage of the grid, $I_i$ is the input current of each branch, $R_s$ is the resistance of the common wire through which all currents flow, $R_x$ is the resistance of the wire connected to the smart speaker, $R_a$ is the internal resistance of the smart speaker. Note that each branch has its line resistance, we have labeled only the line resistance $R_x$ used in Equation (1). The relationship between the input current of the smart speaker, the source voltage, and the current of each branch is as follows:

$$V_s = (\sum_{i=0}^{n} I_i + I_x)R_s + I_x(R_a + R_x)$$

$$I_x = \frac{V_s}{R_s + R_x + R_a} - \frac{R_s}{R_s + R_x + R_a}\sum_{i=0}^{n} I_i \tag{1}$$

Obviously, the input current of the smart speaker $I_x$ is influenced by the current of the other branch $I_i$. As mentioned in Section 3.2, PFC signals (feedback current) is part of $I_i$. When the input current of the smart speaker $I_x$ is influenced by $I_i$, it is influenced by the PFC signals generated by other branches.

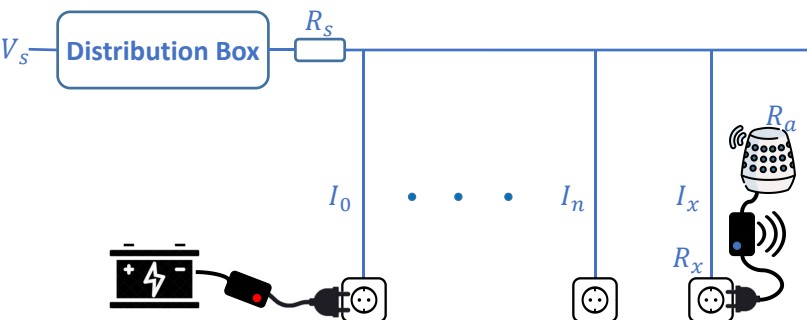

**Figure 5.** Illustration of the household grid.

### 3.4. Transmitted PFC Signals Can Drive Power Supply to Generate Sounds

This section explains how $I_x$, which contains the PFC signals, causes the power supply to emit sound. A typical power supply contains multiple capacitors and inductors inside. When current flows through the power supply, these capacitors and inductors generate sound signals under the action of the current. For example, Figure 6 shows a typical inductor, which consists of a magnetic core and a coil wound on it. A changing current produces a changing magnetic field, and the core is repeatedly stretched in the direction of magnetization to produce sound, a phenomenon known as magnetostriction [46]. Figure 7 shows a capacitor. When a high-frequency current acts on the capacitor, the capacitor deforms. This phenomenon is called the inverse piezoelectric effect. The deformation of the capacitor acts on the circuit board to produce sound. In a preliminary study, we also observed that the sound generated by the power supply is consistent with the PFC signals in terms of features (see Section 4.3).

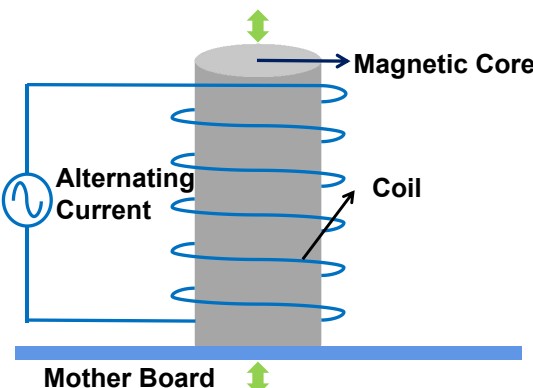

**Figure 6.** Magnetostriction of magnetic core under influence of alternating current.

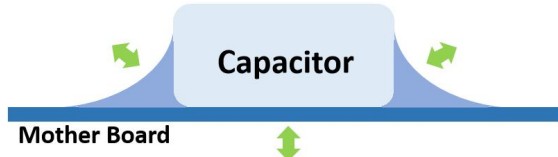

**Figure 7.** Capacitor exhibits inverse piezoelectric effect due to electric field.

## 4. Preliminary Study

In Section 3, we showed that e-bike charging can generate a special PFC current which can drive the power supply in the grid to generate sound from a distance. Before we can use this principle for e-bike charging health monitoring, we need to verify the following inferences: (1) The differences between the PFC currents are sufficient to distinguish each other. (2) The PFC currents is time invariant. (3) The transmitted PFC currents can generate sounds of the same frequency.

We conducted a preliminary study in two rooms (rooms A and B, see Figure 1) under a household grid to verify these inferences. In Room A, we connected different appliances or e-bikes to the grid (listed in Tables 1 and 2, correspondingly) In Room B (see Figure 8), we connected a Tmall smart speaker to the grid and used a sound card with a maximum sampling rate of 192 kHz to receive the sound signal from the smart speaker's power supply. At the same time, we captured the PFC signal (current) with an ACS712 current sensor and Analog Discovery2 (AD2).

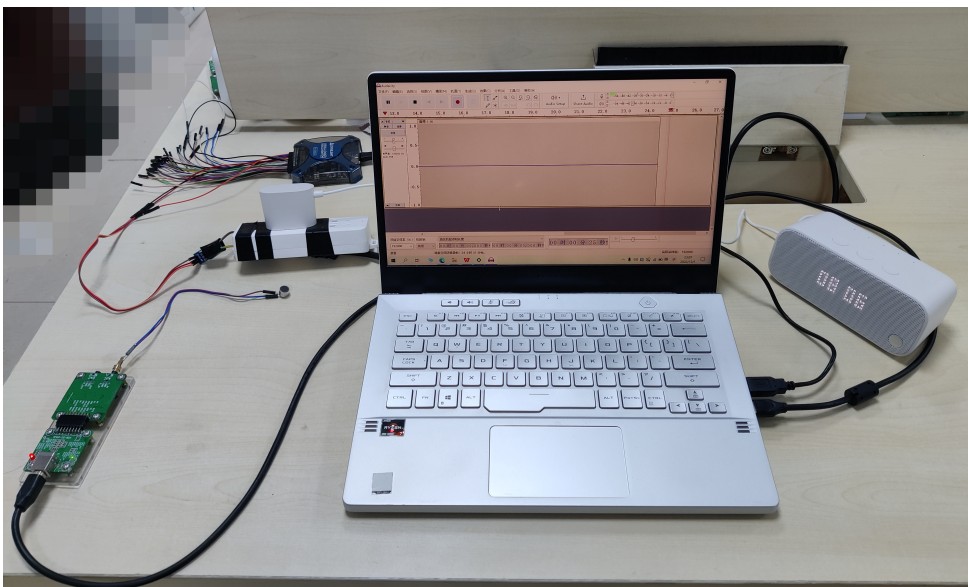

**Figure 8.** Illustration of data collection equipment.

**Table 1.** E-bikes used in preliminary study (marked with *) and evaluation.

| Brand | Model | Battery Type | Power Rating (W) |
|---|---|---|---|
| YADEA (a) * | DE2 | Lithium | 180 |
| YADEA (b) | TDT1241Z | Lithium | 160 |
| AIMA * | D260TZA-L4812 | Lithium | 110 |
| NIU * | UQis | Lithium | 160 |
| FOREVER * | Sport | Lead-acid | 120 |
| XDAO * | XiaoK | Lead-acid | 160 |
| PALLA | K11 | Lithium | 210 |
| SUNRA | TDT4960Z | Lead-acid | 110 |
| AiSUN | LOLLIPOP | Lead-acid | 120 |
| Lvliang | TDR183Z | lead-acid | 175 |

**Table 2.** Appliances used in preliminary study and evaluation.

| Type | Brand-Number (Power Rating) |
|---|---|
| Light | Xiaomi-2 (9 W, 9 W); PHILIPS-1 (18 W) |
| Projector | Sony-1 (225 W) |
| Monitor | PHILIPS-1 (21 W); Dell-1 (50 W) |
| Laptop | Hp-2 (65 W, 90 W); Lenovo-1 (65 W) |
| Phone charger | Huawei-1 (40 W); Xiaomi-2 (33 W, 35 W) |

### 4.1. Difference

To verify Inference 1, we used AD2 to collect multiple data segments at the different charging stages of each e-bike, with each lasting for 20 s. Similarly, we also collected the PFC signals of the remaining 12 electrical appliances. We performed 2D correlation analysis on the spectrograms of these data [47]. Figure 9 shows the cumulative distribution function (CDF) of the correlations. The correlation between different e-bikes and different appliances does not exceed 0.69, and the correlation between different charging stages of the same e-bike does not exceed 0.77. Experiments have shown that the differences between PFC signals are sufficient to distinguish the different appliances and e-bikes charging stages.

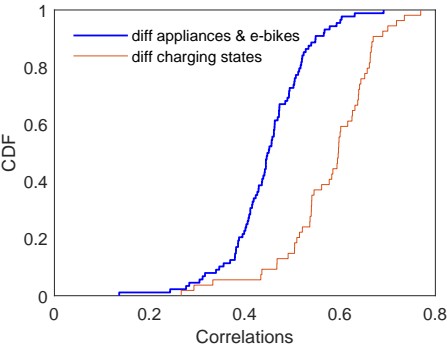

**Figure 9.** Blue is the CDF of 2D correlation between different appliances and e-bikes (including different charging stages). Orange is the CDF of 2D correlation between different charging states of the same e-bike.

### 4.2. Invariance

To verify Inference 2, for each e-bike, a set of data was collected every 5 days using AD2, with each set containing multiple data segments from three charging stages and each segment lasting 20 s, and we collected such data over 1 month. We then performed a 2D correlation analysis between the same charging states of the same charger by using the first set of data as a benchmark. We averaged the correlations for the different charging stages of the same e-bike and also averaged the correlations for other appliances. Figure 10 shows that the correlation between PFC signals can always be maintained above 0.94 within a month, which suggests that the PFC signals remained stable over time.

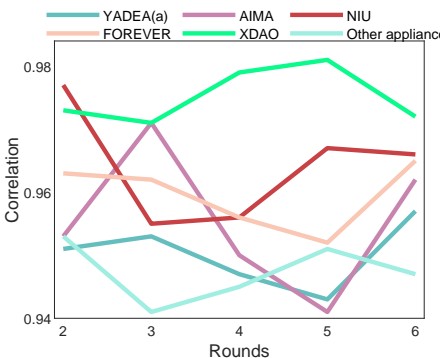

**Figure 10.** 2D correlation of 5 rounds over period of 30 days.

### 4.3. Consistency

To verify Inference 3, in Room A, we connected an HP laptop to the grid. Then, in Room B, we simultaneously measured the sound signal generated by the power supply and the PFC signal (current). Figure 11 shows the power spectrum of the current and the corresponding sound signal, both of which have identical frequency spikes at 21.28 kHz. To better present the information in the power spectrum, we took the logarithm of the vertical axis in conjunction with the definition of decibels ($N_{dB} = 10 \lg(P_x/P_0)$, and $P_0$ is the custom reference value); therefore, the vertical axis unit in Figure 11 is dB. This shows that the sound signal generated by the power supply is characteristically consistent with the corresponding current signal.

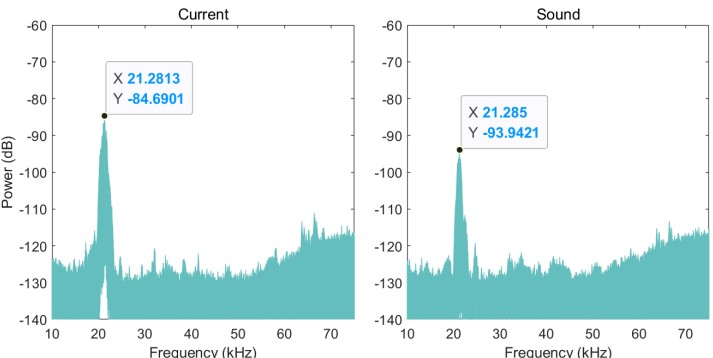

**Figure 11.** Spectrum of simultaneously collected current and sound signal.

## 5. System Design

As shown in Figure 12, the proposed system comprises two phases: Registration Phase and Health Monitoring Phase.

Each e-bike needs to be registered individually. During the registration phase, the e-bike must undergo a complete charging cycle, during which the system continuously collects the sound signals generated by the power supply. Because the background sound can be collected before the e-bike is connected to power in the registration phase, the registration phase can use spectral subtraction to weaken the background noise and the interference from other appliances' PFC signals. Combined with VMD and periodicity detection, we can obtain a signal containing only information about the e-bike charging stage. This signal will be used to generate the charging stage template. The system compares the template information obtained at different times to determine the current charging stage and calculates the duration of each charging stage, thereby generating the normal charging pattern of the corresponding e-bike. The system can register different models of e-bikes in the market in advance for users' convenience.

In the health monitoring phase, the system first preprocesses to remove background noise and then obtains a sound signal that may contain the PFC features of multiple appliances and e-bikes. By matching the template information obtained in the registration

phase, the system can determine which e-bikes are charging in the grid and what charging stage they are in. The system can obtain the duration of the constant current and constant voltage stages from the charging stage information collected many times. After comparing it with the normal charging mode, the system can judge whether these two stages' durations are normal to analyze the e-bike's charging health.

In the following sections, we will introduce these phases in detail.

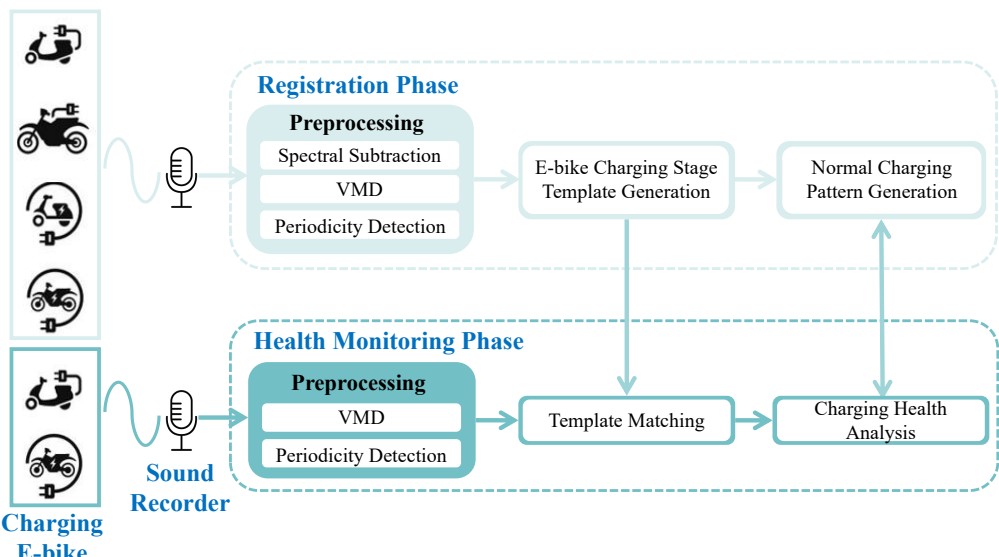

**Figure 12.** System overview of *SingMonitor*.

## 6. Methodology

In this section, we will introduce the methods used by the system.

### *6.1. The Registration Phase*

#### 6.1.1. Spectral Subtraction

To generate a better e-bike charging stage template, our first step is to improve the signal-to-noise ratio. Because the noise in the system changes relatively slowly and the target signals generated by other appliances can remain stable, it is well suited for processing the signal by spectral subtraction [48] to obtain cleaner template information. The sound signal $S_r$ received by the microphone can be expressed as follows:

$$S_r = S_{pfc}(\omega_0) + N_{bg+opfc} \tag{2}$$

where $S_{pfc}(\omega_0)$ refers to the sound signal generated by the influence of the e-bike PFC signal, and $\omega_0$ is the central frequency of the signal. $N_{bg+opfc}$ is the background noise (including ambient noise, the internal noise of the power supply, and the sound card's noise floor) and interference from other appliances' PFC signals. $N_{bg+opfc}$ is time-invariant and follows a specific distribution. So we can attenuate $N_{bg+opfc}$ with spectral subtraction:

$$|\hat{S}_{pfc}|^2 = \begin{cases} |S_r|^2 - \alpha \times E[|N_{bg+opfc}|] & |S_r|^2 \geq \alpha \times E[|N_{bg+opfc}|] \\ \beta|\hat{S}_r|^2 & otherwise \end{cases} \tag{3}$$

where $a$ is the over-subtraction factor; $b$ is the gain compensation factor; $E[|N_{bg+opfc}|]$ represents the estimated background noise and interference; The estimation of the PFC signal can be obtained by performing the inverse fast Fourier transform of $|\hat{S}_{pfc}|^2$.

#### 6.1.2. Variational Mode Decomposition

We further processed the signal to extract PFC features. Because the center frequency of the PFC signals can remain stable, we can use Variational Mode Decomposition (VMD)

to further process the data. VMD can decompose the estimate of the PFC signal ($\hat{S}_{pfc}$) obtained by spectral subtraction into *K* narrow band components ($u_k$) with center frequency ($\omega_k$), i.e., intrinsic mode functions (IMFs).

We first obtain the marginal spectrum of the IMFs using the Hilbert transform; then, we modulate the spectrum of each IMF to the corresponding base band by mixing the center frequencies; and finally, we estimate the bandwidth using Gaussian smoothing. To ensure that the sum of the IMFs obtained from the decomposition is equal to the target signal, we added a restriction: $\sum_k u_k = \hat{S}_{pfc}$. So the objective function and constraints for applying VMD to the PFC signal can be expressed as follows:

$$\min_{u_k, w_k} \{|| \sum_k u_k(t) - \hat{S}_{pfc}||_2^2 + \alpha \sum_k ||\partial(t)[u_k(t)e^{-jw_k t}]||_2^2\} \tag{4}$$

$$s.t. \sum_k u_k = \hat{S}_{pfc} \tag{5}$$

where $\alpha$ is the penalty factor. The Lagrange multiplier method can transform the above optimization problem into an unconstrained one and guarantee accuracy when reconstructing the original signal, which is followed by the Alternating Direction Method of Multipliers algorithm to locate the saddle point [49].

### 6.1.3. Periodicity Detection

The collected sound signals are often mixed with background noise like human voices and device operating sounds, which challenge generating charging status templates. Worse, these noise signals are distributed on a broader frequency band after VMD decomposition; therefore, band-pass filtering cannot remove them. We noticed that the background noise is an aperiodic random signal, however, the PFC feature is periodic, which allows us to reduce noise by periodicity detection further.

To determine the periodicity of IMFs, we first calculated the upper envelope of each IMF and then derived the corresponding auto-correlation coefficient. Then, we divide each IMF into segments, where the auto-correlation coefficient's peak determines each segment's starting point. The length of each segment should exceed the fundamental period of the corresponding IMF, which we set empirically to 0.05 s. Finally, we computed the Pearson Correlation Coefficient (PCC) between segments, and we considered an IMF as periodic when its average PCC reaches a set threshold.

### 6.1.4. Template Generation

After preprocessing, we obtained clean sound signals. For each charging stage *k* of the e-bike *l*, the corresponding processed sound signal is denoted as signal $S_{kl}$. We used the short-time Fourier transform (STFT) with a Hanning window (4096 points) with a shifting interval of 441 points. For window *m*, the corresponding spectrogram is denoted as $S'_{kl,m}$. We then generated a template for real-time traces to match with. The process can be expressed as follows:

$$T_{kl} = \frac{1}{M} \sum_{m=1}^{M} (S'_{kl,m}) \tag{6}$$

where $T_{kl}$ is the generated template and *M* is the number of windows. In this paper, the time length *M* of the templates is set to 200 frames. The templates are then preserved for matching real-time collected sounds.

### 6.1.5. Normal Charging Pattern Generation

The system records when the templates are generated and compares the templates obtained at different times to obtain the duration of each charging stage (*SingMonitor* is mainly concerned with the duration of the constant current and constant voltage stage). The system uses the above information to build the normal charging mode of the corresponding e-bike, which will be used for charging health analysis in the health monitoring phase.

### 6.2. The Health Monitoring Phase

In the registration phase, we obtained templates for different charging stages of various e-bikes under laboratory conditions and built the normal charging pattern of the corresponding e-bike. In this section, we introduce the preprocessing of sound data collected by the user, the template matching method, and how to use the normal charging pattern to judge the charging health.

#### 6.2.1. Preprocessing

As in the registration phase, the sound data captured by the user also contains noise and, therefore, needs to be processed for noise reduction. However, unlike the registration phase. We do not require users to collect background sounds in advance. Therefore, only the VMD and periodicity detection will be used for noise reduction in the registration phase. $S'_r$ represents the sound signal captured directly by the user, and the corresponding sound signal after preprocessing is $\hat{S}_{pfc}(\omega_0)'$.

#### 6.2.2. Template Matching

In developing this metric, we considered two factors in combination.

**The center frequency similarity:** The center frequency varies significantly between different appliances and e-bikes (as mentioned in Section 3); Therefore, we developed the measure $D1$. It considers the difference between the center frequencies of the target and the template, measured using the Euclidean distance. Firstly, since different frequency components in the template have different impacts on the matching, we construct a weight for each frequency component, denoted as $\omega_1$. The weight function $\omega_1$ is defined as:

$$\omega_1 = F_D(f)T_{kl}(f) \tag{7}$$

where $T_{kl}$ is the template and $F_D$ is a filter function. Then, the distance can be determined by integrating local distance $\gamma_i$ in the time-frequency domain, weighted by $\omega_1$, which can be expressed as:

$$D1_i = \sum_{t=1}^{10} \sum_{f=1}^{2048} \omega_1 \gamma_i(t, f) \tag{8}$$

**The frequency width similarity:** During the experiments, we found that the similarity of the central frequency is not the only factor that influences the template matching procedure. For example, the difference between the different charging states of the same e-bike is more obviously in bandwidth. To measure this metric, similarly, we developed the evaluation metric $D2$, which is expressed as follows:

$$D2_i = \sum_{t=1}^{10} \sum_{f=1}^{2048} \omega_2 width_i(t, f) \tag{9}$$

where $width_i$ is the frequency width of each frequency band, defined as the frequency components that are larger than $0.2 \times f_{max(i)}$, and $f_{max(i)}$ is the max strength of the center frequency of this frequency band. Ten frames (frame is the window size in the STFT) of data are used for template matching.

We then combined these two metrics to measure the similarity of the template and the actually acquired signal:

$$D_i = D1_i + \lambda D2_i \tag{10}$$

where $\lambda$ is the penalty factor that ranges from 0 to 1. $\lambda$ is tuned in the registration phase and applied in the monitoring phase.

#### 6.2.3. Charging Health Analysis

When the system starts, the duration of all charging states is set to 0. The system collects sounds at time interval $\Delta T$, obtains which e-bikes are charging and their charging

stages by template matching, and then adds $\Delta T$ to the duration of the corresponding charging stage. If a charging state has not been detected for a long time, its duration is reset to 0.

The system compares the duration of the constant current and constant voltage stage with the normal charging pattern of the corresponding e-bike. Once the duration of the constant current and constant voltage phase significantly exceeds the duration in the normal charging pattern (in the actual experiment, the threshold is set to 30%), the system considers that there is a fault with the e-bike battery or charger and provides a warning about its charging health. The system will use the data measured during the health monitoring phase to adjust the threshold of the corresponding e-bike.

## 7. Evaluation

### 7.1. Experiment Setup

#### 7.1.1. Experiment Setup

As shown in Figure 1, We conducted experiments in two rooms under a distribution box. The distance between the sockets in the two rooms is 9m. At the receiving end (room B), We used the Tmall smart speaker's power supply to generate the sound signal. Since the smart speaker's internal data is unavailable, we chose a sound card (connected to an electret condenser microphone) with a maximum sampling rate of 192 kHz to capture the sound signal (see Figure 8). At the PFC signals transmitting side (room A), We used a total of 10 e-bikes and 12 other appliances for the experiment (see Tables 1 and 2). Furthermore, we used an ACS712 current sensor to obtain the charging stage of the e-bike and used it as the ground truth.

#### 7.1.2. Dataset Preparation

A total of 3 datasets were generated in the experiments for micro benchmark, overall evaluation, and robustness evaluation, correspondingly.

**Registration Dataset**: For the register phase, we used 10 e-bikes in the experiments. We conducted multiple 30 s data collections for each charging stage of each e-bike, with a total data length of 90 min, of which we chose the first 20 s to generate templates and the last 10 s to perform template matching and obtain a micro-benchmark of the system.

**Overall Dataset**: For the health-monitoring phase, we conducted a real-world experiment by randomly connecting multiple e-bikes and other appliances in Room A (up to three e-bikes and four other appliances working simultaneously). To reduce the system complexity, we collect 0.5 s of sounds every 15 min (as mentioned in Section 6.2.2, 0.25 s of data is enough for the system to perform a template match). We experimented for 25 days, and the total duration of data collected during the experiment is 381 s. These data were used to evaluate the overall system performance (see Section 7.3).

**Robustness Dataset**: Similarly, to evaluate the robustness of *SingMonitor* (See Section 7.4), five e-bikes were used in the experiment (e-bikes marked with * in Table 1). we experimented on each e-bike under different influencing factors and averaged the results to measure the system's performance. The corresponding total data length is 620s.

#### 7.1.3. Evaluation Metric

As mentioned in Section 3.1, *SingMonitor* mainly uses the constant current/voltage stage duration to determine whether the e-bike is in a healthy charging status. Therefore, we mainly evaluate the accuracy of the system for classifying the different charging stages of different e-bikes. $F1_i$ score is used to assess each class's performance, and micro $F1$ score is used to measure overall system performance. $F1_i$ can be computed as follows:

$$F1_i = \frac{2 \cdot p_i \cdot r_i}{p_i + r_i} \tag{11}$$

where $p_i$ and $r_i$ are the precision and recall for a particular charging state of an e-bike, respectively. Essentially, $F1_i$ score and micro $F1$ score are positively correlated with the sys-

tem's classification performance. In addition, we evaluated the overall system performance in Section 7.3 by counting the duration of the different charging phases.

### 7.2. Micro-Benchmark

This section evaluates how different parameters affect the *SingMonitor* performance. The Registration Dataset is used for evaluation.

#### 7.2.1. STFT Window Size

The window size in STFT determines the resolution of the spectrogram. The time resolution of the spectrogram decreases as the window size increases, while the frequency resolution does the opposite. The time-frequency resolution will affect the template generation and matching, thereby affecting the system performance. We experimented to select the best window size. Figure 13a shows the micro F1 score for different window size, we finally choose 4096 (micro F1 score is 0.97) as the window size for STFT.

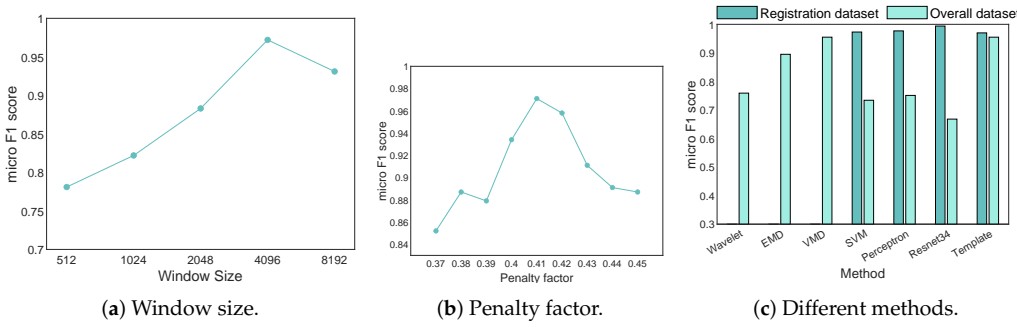

(**a**) Window size.          (**b**) Penalty factor.          (**c**) Different methods.

**Figure 13.** Performance of *SingMonitor* with different micro-benchmark.

#### 7.2.2. Penalty Factor $\lambda$

The penalty factor (see Section 6.2.2) is used to adjust the weighting of the central frequency similarity to the frequency width similarity. We experimented to determine the size of this value. Figure 13b shows that the system performed best when the penalty factor is 0.41 (micro F1 score is 0.97), so we used 0.41 as the penalty factor in all subsequent experiments.

#### 7.2.3. Different Methods

Different noise reduction schemes and different classification methods will have an impact on system performance. We changed the preprocessing method to wavelet transform or empirical mode decomposition (EMD) combing wavelet threshold, which we tested using the overall dataset. Figure 13c shows that the variational mode decomposition (VMD) method performs best. We also used the SVM, multi-layer perceptron, and resnet-34 model to classify e-bikes and their charging stages. The registration dataset is used to train and test these models, and the overall dataset is used for testing only. As shown in Figure 13c, in the registration dataset, all of these models perform well. However, because the overall dataset is interfered by the PFC signals generated by other appliances, the performance of these models on the overall dataset degrades to varying degrees. The template matching is designed based on the characteristic that the center frequency and bandwidth of PFC signals generated by different appliances and e-bikes' different charging stages are different. So it is more resistant to interference and thus performs more stably.

### 7.3. Overall Performance

In this section, we evaluate the overall performance of *SingMonitor*. The Overall Dataset is used for evaluation.

### 7.3.1. Template Matching Evaluation

As mentioned in the evaluation procedure, during the experiment, the system collected a 0.5 s sound signal every 15 min and performed a template matching to determine which e-bikes were charging in the grid and what charging stage they were in. Figure 14a shows that F1 score can reach above 0.94 for the different charging stages of different e-bikes. This proves that the system can achieve high performance in a complex environment with multiple e-bikes and other appliances.

### 7.3.2. Duration of Charging Stages

The system calculates the durations of different charging stages and compares them with the normal charging pattern of the corresponding e-bike to judge charging health. During the experiments, the duration of the constant current and constant voltage stages was within the normal range for all 10 e-bikes, and the system judged the charging health of these e-bikes to be fine. As shown in Figure 14b, we cumulated the duration of the different phases of each e-bikes and compared with the real time (the system does not care about the duration of the float stage, but it can also be calculated). The error between the system calculation time and the real time does not exceed 9%. The time calculated by the system is high compared to the real time because the system's misjudgment of the charging stage tends to increase the charging time.

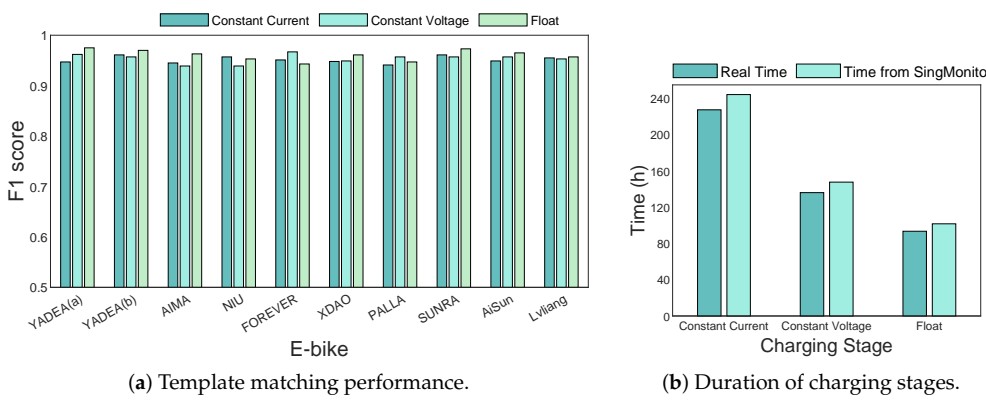

(**a**) Template matching performance.  (**b**) Duration of charging stages.

**Figure 14.** Overall performance of *SingMonitor*.

### *7.4. Robustness*

In this section, we evaluate the robustness of *SingMonitor* under different influences. The Robustness Dataset is used for evaluation.

### 7.4.1. Impact of Number of Appliances

We gradually increased the number of other types of appliances in the grid (1∼10) to evaluate the system. Figure 15a shows that, as the number of appliances increases, the system's performance decreases. This is because the increase in the number of appliances made the PFC signal generated by e-bikes more susceptible to interference in the frequency domain, which affects template matching. When the number of appliances does not exceed 6, the micro F1 score can be maintained above 0.9.

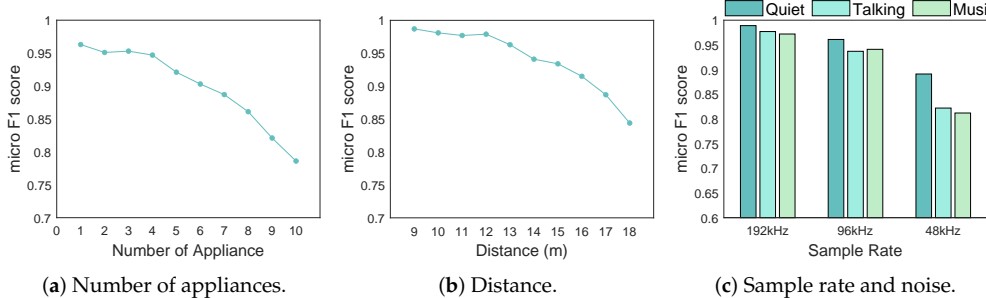

**Figure 15.** Robust performance of *SingMonitor*.

### 7.4.2. Impact of Distance

We increased the propagation distance of the PFC signals by extending the wires. As shown in Figure 15b, the system performance decreases with increasing distance because the strength of the PFC signal will gradually attenuate as the propagation distance increases. However, micro F1 score can still be maintained above 0.91 in the range of 16 m.

### 7.4.3. Impact of Sound Sampling Rate and Environmental Noise

Aliasing errors occur when the sampling rate does not meet the Nyquist sampling theorem. We varied the sample rate (192 kHz, 96 kHz, 48 kHz) of the sound card and conducted experiments with different environmental noises (quiet, talking, playing music) to verify the effects of the sample rate and noise on the system. Figure 15c shows that the system's performance deteriorates as the sampling rate decreases and the noise increases. Because of the aliasing effect, we can extract distorted PFC features at low sampling rates, so the system can still work at low sampling rates (the micro F1 score at 48 kHz is 0.89). At low sampling rates, the effect of noise on the system is more pronounced because the noise cannot be completely removed, and its frequency is lower, which is more likely to interfere with the distorted PFC features.

## 8. Discussion

*SingMonitor* is limited by several issues.

First, the microphones of mobile devices, such as smartphones, often use low-pass filtering, i.e., an anti-alias filter (AAF), before sampling the audio signals. Due to the AAF, mobile devices may not be able to capture clear PFC frequencies, thus affecting *SingMonitor* performance. However, existing studies attempted to recover the signals that underwent AAF processing by methods such as non-linearity [50], and we will subsequently try these methods.

Second, the center frequency of some electrical appliances is close to that of e-bikes, which may affect the monitoring of e-bikes when these devices are in use. In particular, some appliances (like the air conditioner, laundry, refrigerator) change the frequency when operating, causing the appliance noise removal method based on spectral subtraction to fail, reducing the system's performance. However, these devices are few in daily life. Besides, we can also use other methods (e.g., low-frequency sound) and the sound of the user using the appliance, as compensation for classification or collect and analyze their signal characteristics in advance to remove their interference to the system.

## 9. Conclusions

We proposed and implemented a system that uses sounds emitted by the power supply to monitor e-bike charging health. Compared with existing fire warning systems or other e-bike monitoring systems, *SingMonitor* uses power supplies everywhere for monitoring, which can be implemented on mobile devices and achieve long-range monitoring.

A noise cancellation scheme combining VMD, periodicity detection, and spectral subtraction is used to cancel background noise and other appliances' interference. We proposed a

template matching based scheme and designed a new distance metric to distinguish different charging stages. Experiments showed that *SingMonitor* could achieve an F1 score of 0.94 in identifying 10 e-bikes' charging stages, with a detection distance of 9m+.

**Author Contributions:** Conceptualization, X.J., Y.L. and Y.-C.C.; Methodology, X.J. and L.Y.; Software, X.J.; Supervision, Y.-C.C. and G.X.; Validation, X.J. and L.Y.; Writing—original draft, X.J. All authors have read and agreed to the published version of the manuscript.

**Funding:** This research received no external funding.

**Institutional Review Board Statement:** Not applicable.

**Informed Consent Statement:** Not applicable.

**Data Availability Statement:** The data involved in this study are available upon reasonable request. The data are not publicly available because they are part of further research.

**Conflicts of Interest:** The authors declare no conflict of interest.

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
