# Peer review of "SingMonitor: E-bike Charging Health Monitoring Using Sound from Power Supplies"

_applsci, doi:10.3390/app13053087_

Round 1

Reviewer 1 Report

The article introduces SingMonitor, a methodology for monitoring the health of e-bike charging. The proposal is novel and is based on identifying the sound signature of the PFC effect of e-bike chargers on power supplies. The article is very well written and formatted. The technical aspect of the paper is very detailed, presenting each stage of its development in a comprehensive and didactic manner. The experimental verification of the proposed technique follows a very solid methodology, and the presented results help to corroborate a good preliminary performance. In general, the article is good and highly relevant to the field. As a result, I recommend that the article be accepted in this journal after minor revisions:

1- The text between lines 31 and 35 is confusing, affecting the presentation of the chosen references.

2- Figures 1, 2, 3, and 4 appear before their first mentions in the text, which is inconvenient for the reader and hinders their reading by forcing them to return. For a better reading experience, the figures should be close to the first point where they were referenced throughout the text.

3- Reference 44 is not formatted properly.

4- Line 489 - "Impact of Distance", mistyped the word Distance

Author Response

Thanks for the valuable comments and revision suggestions.

We have adjusted the content of lines 31 to 35 as you suggested and repositioned the images to improve the reader's reading experience. We have fixed the formatting of reference 44 and the spelling error you mentioned.

To track the changes, you can download our manuscript source code and uncomment line 66 of template.tex.

Reviewer 2 Report

The idea of the work is extremely simple and can be obtained by many other means. It is not suitable to be published as a yechnical paper.

The idea is worth a graduation project in the last year of Engineering school.

Author Response

Thanks for the valuable comment. 
Firstly, we believe it is a valuable issue to research e-bike charging health because the fire accidents caused by e-bike charging (nearly 18,000 occurrences in China alone in 2021) result in significant yearly property damage and casualties.

Secondly, although the existing fire alarm systems can reduce the losses caused by e-bike charging fires, these methods have high installation and maintenance costs and only work after a fire occurs. Load monitoring methods for appliance detection can also monitor the charging status of e-bikes, but they require extra hardware such as power meters or ammeters. Some side-channel methods place sensors near the appliances and can monitor the e-bike charging status using magnetic or acoustic signals generated when the appliance is running. However, the monitoring range of these methods is limited. This paper uses the phenomenon that PFC signals can drive the power supply to generate sound, and propose the first feasible solution to monitor the e-bike charging health using the sound at a relatively long distance.

And we believe the readers of MPDI Applied Sciences will be highly interested in the paper because it can monitor the e-bike charging health by sound at a long distance (9m+).

Reviewer 3 Report

Thanks authors for interesting contents of the submitted paper.

The reviewer would like to ask a few questions and request some clarifications as follows:

1) Can the proposed method provides multiple e-bike charging cases? If yes, please provide description with proper evidence

2) Feasible distance for the proposed method is 9 meter between one outlet to the other. What is the line resistance or impedance for this 9 meter wire?

3) What are the power ratings of devices in Fig. 4?

4) What about in case of the proposed device with home appliances such as laundry machines,  refrigerators, and electric vehicles?

5) In Fig. 5, what are the values of Rs, Rx, and Ra? In addition, are there any line resistance in between  I1, I2, ... In?

6) In Fig. 8, what CDF stands for?

7) In Fig. 10, the unit of current should be A or mA not dB?

8) In Table.1 and 2, please add power rating for each device

9) Subtitle 7.4.2, correct typo: istance --> distance

10) Provide a system block diagram for the overall data process  

Author Response

Thank you for your valuable revision suggestions. The following is our response and revisions.

  1. Can the proposed method provide multiple e-bike charging cases? If yes, please provide description with proper evidence.

    revision: The overall dataset we mentioned in Sec 7.1.2 contains multiple e-bikes charging cases. In collecting the overall dataset, we simultaneously connect multiple e-bikes and appliances within the grid and use the system to collect sound signals to determine which e-bikes are charging and what charging stage they are in. We have placed the evaluation results in Sec 7.3. Because of the arrangement of the article structure, we did not put the dataset with the results.
  2. Feasible distance for the proposed method is 9 meter between one outlet to the other. What is the line resistance or impedance for this 9 meter wire?

    revision: According to national standards, the resistance of copper conductor wire with a cross-sectional area of 2.5mm2 is no higher than 7.98 Ω/km, so the line resistance of this 9-meter wire should be below 0.071Ω.

  3. What are the power ratings of devices in Fig. 4?

    revision: We have revised Fig. 4 to add the power ratings of devices.
  4. What about in case of the proposed device with home appliances such as laundry machines, refrigerators, and electric vehicles?

    revision: We discussed the impact of these appliances in Sec 8. The frequencies generated by these appliances can change over a considerable range during operation. So SingMonitor's performance is affected when these appliances are present. However, the number of such household appliances is small, and we can consider collecting and analyzing their signal characteristics in advance to remove their interference with the system.

  5. In Fig. 5, what are the values of Rs, Rx, and Ra? In addition, are there any line resistance in between I1, I2, ... In? 

    revision: In Sec 3.3, we want to show that the PFC signal (it is part of the corresponding branch's current Ii, i = 1,2, ... n) generated in one branch of the home grid can propagate along the wire and affect the current signal in other branches.
    Fig. 5 is an abstract illustration, which is used to derive the equation to demonstrate that the branch current Ix is influenced by the other branch currents (I1, I2, ... In) and, thus, by the PFC signals generated by the other branches.
    Because Fig. 5 is an abstract illustration, there are no specific values for Rs, Rx, and Ra. This paper is also not concerned with the value of the resistance in the grid.
    Similarly, there are line resistances in the branches (I1, I2, ... In). However, the derivation of Equation 1 did not use the line resistances of these branches, so Fig. 5 does not identify these line resistances with specific symbols.
    We have revised the statements in Sec 3.3 to be less misleading.
  6.  In Fig. 8, what CDF stands for?

    revision: We did not give the CDF's full name in the manuscript, and we have fixed this bug in the latest version (see Sec 4.1).
    CDF is an abbreviation for the cumulative distribution function. In this paper, we use CDF plots to show the probability distribution of the 2D correlation between PFC signals.

  7. In Fig. 10, the unit of current should be A or mA not dB?

    revision: The unit of the collected current signal is ampere. In order to study whether the frequency characteristics of the current signal and the sound signal are consistent, we analyzes the frequency characteristics by calculating the power spectrum of signals.
    The power spectrum of the current signal has a horizontal coordinate in Hz and the vertical coordinate in A2/hz.
    In order to better present the information in the power spectrum, it is common practice to take the logarithm of the vertical axis in conjunction with the definition of decibels (NdB = 10lg(Px / P0), P0 is the custom reference value in A2/hz), so the vertical axis unit in Fig. 10 is dB.
    We have added corresponding explanations in Sec 4.3 to help readers understand.

  8. In Table.1 and 2, please add power rating for each device。

    revision: We have added power ratings for the devices in Table. 1 and 2 as you suggested.

  9. Subtitle 7.4.2, correct typo: istance --> distance

    revision: We have fixed this typo error.

  10. Provide a system block diagram for the overall data process.

    revision: We place Fig. 12 in Sec 5 to illustrate the overall data process. 
    The data processing in the system is divided into two main parts: Registration Phase and Health Monitoring Phase.
    During the registration phase, the system preprocesses the collected sound signal, then uses the processed data to generate charging stage templates, and then calculates the charging time of different stages to generate the normal charging pattern of the corresponding e-bike.
    During the health monitoring phase, the collected sound signals also need to be preprocessed first, after which the system combines the template information obtained in the registration phase and the template matching algorithm to determine which e-bikes are charging and their charging duration. Finally, the system compares this information with the normal charging pattern to analyze the e-bike charging health.

Round 2

Reviewer 2 Report

Dear authors;

Unfortunately, I am not still convinced enough with the application of your sensor and its importance to your application.

Many other simple techniques can be employed for the purpose of your target.

Author Response

Thank you for your suggestions. We did not make some details clear enough before and thus misleading you. However, we did encounter many challenges in our research, the main ones being the following:

  1. How to find a method to monitor the e-bike charging health from a distance without extra hardware?

    Based on the fact that the power factor correction (PFC) circuit of the e-bike can generate different feature current signals under different charging stages, and the phenomenon that the power supply produces sound under the action of current, we designed a side channel monitoring system using sound for the e-bike charging health monitoring, with a monitoring distance of 9m+.

  2. How to remove the interference of background noise from the power grid, internal components of the power adapter, and sound collection devices?

    We used the wavelet transform or empirical mode decomposition (EMD) combining wavelet threshold to preprocess the signal. However, wavelet transform is more suitable for dealing with abrupt signals, while the noise signal in this paper changes relatively slowly, so the system performs unfavorably when using wavelet transform for noise reduction. The EMD combined with the wavelet threshold method also cannot match the signal characteristics of this paper well and, therefore, cannot achieve the best noise reduction effect. Finally, considering that the PFC signals are periodic, their center frequency is stable, and the noise signals are aperiodic, we used the variational modal decomposition combing the period detection method for preprocessing.

  3. To improve the system's usability, the data collected in the registration phase (Registration dataset: divided into training set and test set, see Sec 7.1) contains only the characteristics of PFC signals generated by e-bikes. However, when analyzing the charging stages of the e-bike in the health monitoring phase, the signals collected by the system (Overall dataset: for testing only) are interfered by PFC signals generated by other appliances. So it is challenging to design a system that can accurately identify the e-bike and its charging stage in a complex appliance environment with limited training data.

    We used the SVM, multi-layer perceptron, and resnet-34 model to classify e-bikes and their charging stages. The Registration dataset is used to train and test these models, and the Overall dataset is used for testing only. In the Registration dataset, all of these models perform well. However, because of other appliances' interference, the performance of these models on the Overall dataset degrades to varying degrees. Considering that the types, numbers, and combinations of appliances in different homes vary, using these models requires the user to collect a large amount of data for training during the registration phase, which can significantly reduce the system's usability. The template matching is designed based on the characteristic that the center frequency and bandwidth of PFC signals generated by different appliances and e-bikes’ different charging stages are different. So it is more resistant to interference and thus performs more stably. Under careful consideration, the system finally uses the template matching algorithm.

We have also revised the article based on your suggestions. We modified the introduction to describe better the challenges and the corresponding solutions. We revised Sec 6 to improve our description of the method. In the evaluation (see Sec 7.2), we provide the impact of different noise reduction schemes and classification methods on the system performance. We also modified related works by adding studies related to battery monitoring [A, B, C, D], which show that battery monitoring has an important impact on the safe use of batteries. [A] measures the battery's voltage and temperature by installing wireless sensors inside each cell to assess the battery's state of health. [B] designs an IoT system to obtain information such as battery voltage and upload it to the control interface to facilitate the user to know and deal with the abnormal status of the battery in time. [C] designs a battery management system using sensors to obtain parameters such as battery voltage, current, and temperature. These parameters are used to train an LSTM model to monitor and protect the battery status. [D] proposes an electrochemical model that monitors the cell state by calculating parameters such as open-circuit voltage, liquid-phase diffusion, etc. The calculation of these parameters relies on the reading of current, voltage, and other data. However, these methods require extra hardware to obtain data, such as current and voltage, which raises the system cost. Compared with these direct-style monitoring methods, this paper proposes a sound-based side-channel method to monitor the e-bike charging health without information such as current and voltage of the battery. Also, these methods are mainly for electric vehicles [A, B] or lithium batteries [C, D], which are not necessarily applicable to our target.

Finally, according to your suggestion, we use MDPI's editing service to improve the writing level of the article.

[A] Schneider, M.; Ilgin, S.; Jegenhorst, N.; Kube, R.; Püttjer, S.; Riemschneider, K.R.; Vollmer, J. Automotive battery monitoring by wireless cell sensors. In Proceedings of the 2012 IEEE International Instrumentation and Measurement Technology Conference Proceedings. IEEE, 2012, pp. 816–820.

[B] Abd Wahab, M.H.; Anuar, N.I.M.; Ambar, R.; Baharum, A.; Shanta, S.; Sulaiman, M.S.; Hanafi, H. IoT-based battery monitoring system for electric vehicle. International Journal of Engineering & Technology 2018, 7, 505–510.

[C] Thomas, J.K.; Crasta, H.R.; Kausthubha, K.; Gowda, C.; Rao, A. Battery monitoring system using machine learning. Journal of Energy Storage 2021, 40, 102741.

[D] Li, J.; Wang, L.; Lyu, C.; Liu, E.; Xing, Y.; Pecht, M. A parameter estimation method for a simplified electrochemical model for Li-ion batteries. Electrochimica Acta 2018, 275, 50–58.

Reviewer 3 Report

Thanks for updating the paper w.r.t. the reviewer's comments.

Author Response

Thank you again for your comments!